# The Complex of Phycobiliproteins, Fucoxanthin, and Krill Oil Ameliorates Obesity through Modulation of Lipid Metabolism and Antioxidants in Obese Rats

**DOI:** 10.3390/nu14224815

**Published:** 2022-11-14

**Authors:** Xi Qiang, Chuanlong Guo, Wenhui Gu, Yuling Song, Yuhong Zhang, Xiangzhong Gong, Lijun Wang, Guangce Wang

**Affiliations:** 1CAS and Shandong Province Key Laboratory of Experimental Marine Biology, Center for Ocean Mega-Science, Institute of Oceanology, Chinese Academy of Sciences, Qingdao 266071, China; 2College of Marine Life Sciences, Ocean University of China, Qingdao 266003, China; 3Nantong Zhong Ke Marine Science and Technology R&D Center, Nantong 226334, China; 4Department of Pharmacy, College of Chemical Engineering, Qingdao University of Science and Technology, Qingdao 266042, China; 5Laboratory for Marine Biology and Biotechnology, Pilot National Laboratory for Marine Science and Technology, Qingdao 266237, China; 6Qingdao Shinan District Center for Disease Control & Prevention, Qingdao 266072, China

**Keywords:** obesity, lipid metabolism, phycobiliproteins, fucoxanthin, krill oil

## Abstract

Phycobiliproteins, fucoxanthin, and krill oil are natural marine products with excellent activities. In the study, we prepared the complex of phycobiliproteins, fucoxanthin, and krill oil (PFK) and assessed the anti-obesity, lipid-lowering, and antioxidant activities in high-fat diet rats. The results showed that the rats significantly and safely reduced body weight gain and regulated serum biochemical parameters at 50 mg/kg phycobiliproteins, 10 mg/kg fucoxanthin, and 100 mg/kg krill oil. Furthermore, the molecular mechanism study suggested that the complex of PFK confined the enzyme activities of lipid synthesis and enhanced antioxidant activity to improve obesity indirectly. The conclusions demonstrated that the complex of PFK has potent anti-obesity and hypolipidemic effects which have potential use as a natural and healthy food and medicine for anti-obesity and lowering blood lipids in the future.

## 1. Introduction

Obesity is caused by an energy imbalance between calorie intake and calorie consumption. According to the World Health Organization, the number of obese people has increased rapidly in recent years. More than 3.4 million adolescents are obese and the age of obese people shows a decreasing trend, which affects the health and development of teenagers [1]. A long-term high-fat diet leads to obesity. Obesity is not only accompanied by chronic diseases such as hyperlipidemia, hypertension, and diabetes [2] but also increased risk of cardiovascular diseases, infections [3], and even increased incidence and mortality of liver cancer, kidney cancer, colorectal cancer, and other cancers [4,5]. Therefore, treating obesity and controlling blood lipids has a beneficial influence on human health. There are many methods to treat obesity, including exercise, dietary changes, surgeries, and drugs. However, exercise and diet take a long time to cause weight loss and it is difficult for severely obese people to lose weight and become less hypolipidemic through exercise and diet. Weight loss surgery may have side effects and unfavorable long-term results [6]. At present, drugs are the most common treatment for weight loss by inhibiting the absorption of fat and promoting bodily metabolism [7]. However, there is a possibility that chemical drugs have side effects bringing hidden risks to health and there is a lack of drugs to achieve simultaneous weight loss and lipid-lowering. Therefore, it is important to find safe, non-toxic, and no side effects causing anti-obesity and lipid-lowering drugs. Some research has shown that some natural products were safe for weight loss and hypolipidemia.

Natural marine products have attracted increasing attention as safe active compounds extracted from marine organisms [8]. Algae are one of the important marine resources. Since ancient times, algae has had medicinal value and health-care functions, providing sources of various biological activity-based natural products. As a widely accepted edible and medicinal algae in China, *Neopyropia yezoensis* has healthcare benefits with lipid-lowering, dissolving phlegm, and so on according to the ‘Compendium of Materia Medica’ [9]. Phycobiliproteins (PBPs) are an important natural active product from *Neopyropia yezoensis*. In a deep study, phycobiliproteins were found to have anti-oxidation, anti-tumor, anti-obesity, anti-inflammatory, and immune-enhancing effects, and fluorescence activities and photosensitivity with wide applications, which have been widely applied in many fields such as food, biomedicine, pharmaceuticals, and environment [10]. There are a few studies about phycobiliproteins and anti-obesity. Mayuko reported that phycocyanin enhanced the adiponectin and thereby elevate the expression levels of aortic endothelial nitric oxide synthase and increased the antioxidant activity, which improved the levels of blood pressure and obesity [11]. In addition, the antioxidant activity could reduce lipid peroxide accumulation and consequently the risk of liver damage. Jin reported that phycocyanin could reduce adipogenesis [12]. The above features laid the basis for deep research into the anti-obesity and lipid-lowering functions of phycobiliproteins.

*Saccharina japonica* is also an important edible and medicinal algae. Due to its health benefits, various bioactive products have been studied in recent years, such as fucoidan, fucoxanthin, and mannitol. Fucoxanthin (FX) is a lipid-soluble carotenoid from brown algae without any side effects [13]. It has been reported that fucoxanthin has many biological properties including antioxidant, anti-obesity, anti-cancer, anti-inflammatory, and photoprotective [14,15]. Fucoxanthin can be used in food and cosmetic industries, biomedicine, and other areas with high value [16]. Recently, a study indicated that fucoxanthin can decrease the expressed activity of adipose triacylglycerol lipase and carnitine palmitoyltransferase, which showed potential candidates to be developed as a new weight-loss drug [17].

Biologically active products are not only from marine algae but also from marine animals. Antarctic krill is a significant marine resource that has been widely studied in recent years. Krill oil (KO) is a natural product from Antarctic krill. Compared with microalgae oil and fish oil, krill oil not only has the same amount of phospholipids, vitamins, flavonoids, minerals, and astaxanthin [18] but also contained high concentrations of eicosapentaenoic acid and flavonoids, which are easily absorbed [19]. Phosphatidylcholine is the most abundant phospholipid in krill oil, which has various biological activities, such as lipid-lowering, anti-inflammatory, cardiovascular disease prevention, neuroprotection, and anti-cancer activities. Krill oil can decrease the activities of cytosolic acetyl-CoA carboxylase and fatty acid synthetase, which inhibited adipogenesis and achieved hypolipidemic [20,21], showing that it is a potent natural product to improve dyslipidemia. Increasing research on the activities of krill oil has gained the attention of food and biomedical applications, which have potential high value.

Now, most research focuses on single natural products playing a role in obesity. However, there are fewer studies about how the complex synergies of several natural products have weight loss and hypolipidemic effects. In this work, we made the PFK complex with three marine natural products: phycobiliprotein, fucoxanthin, and krill oil. It explored the different doses of the complex of phycobiliprotein, fucoxanthin, and krill oil to investigate its ameliorative effect and mechanism on weight loss and hypolipidemia in rat models.

## 2. Materials and Methods

### 2.1. Materials and Chemicals

*Neopyropia yezoensis* was collected from Lianyungang, Jiangsu Province, China. The PBPs were extracted from *Neopyropia yezoensis* as described in a previous study by Wang et al. [22]. The purity of extracted PBPs was 0.97. The FX (purity ≥ 91%) was purchased from the Shandong Jiejing Group Corporation (Rizhao, China), which was extracted from *Saccharina japonica*. The KO (0.57 g/g phospholipids, 0.16 g/g EPA, 0.09 g/g DHA, and 0.25 mg/g astaxanthin) was purchased from Luhua Biomarine (Shandong) Co., Ltd. (Jinan, China). All other chemicals were supplied from local chemical suppliers and were analytical reagent grade unless specifically indicated.

### 2.2. Antioxidant Activities In Vitro

The total antioxidant activity was measured with reference to a test kit (Nanjing Jiancheng Corp., Nanjing, China). A standard curve was developed with absorbance as the vertical coordinate and standard sample concentration as the horizontal coordinate. The reaction mixture contained 5 μL of the sample (7.5 mg/mL PBPs, 1.5 mg/mL FX, 15 mg/mL KO, and 7.5 mg/mL PBPs, 1.5 mg/mL FX, 15 mg/mL KO low dose of PFK complex) and 180 μL of the ferric reducing ability of plasma working solution. After incubation at 37 °C in a water bath for 10 min, the absorbance was determined at 593 nm. The total antioxidant activity of the sample was calculated using the standard curve.

The hydroxyl-free radical scavenging activity of samples was carried out based on the method by Zhou with some minor modifications [23]. The reaction mixtures contained 1 mL sample, 1 mL 9 mmol/L FeSO_4_, 1 mL 9 mmol/L salicylic acid ethanol solution, and 1 mL 8.8 mmol/L H_2_O_2_. The 1 mL of sample was mixed with 1 mL 9 mmol/L FeSO_4_, 1 mL 9 mmol/L salicylic acid ethanol solution, and 1 mL deionized water. Then, the 1 mL deionized water was mixed with 1 mL 9 mmol/L FeSO_4_, 1 mL 9 mmol/L salicylic acid ethanol solution, and 1 mL 8.8 mmol/L H_2_O_2_. These were incubated for 1.5 h at 37 °C in a water bath, the absorbance was measured at 536 nm. The hydroxyl-free radical scavenging activity of samples was calculated using the following formula: hydroxyl-free radical scavenging rate (%) = [A_0_ − (A_x_ − A_x0_)]/A_0_ where A_x_ was the absorbance of sample and H_2_O_2_, A_x0_ was the absorbance of sample and deionized water, and A_0_ was the absorbance of deionized water and H_2_O_2_.

### 2.3. Animal Experiments

All animal experimental procedures in this study were approved by the Qingdao University of Science and Technology Ethics Committee for Animal Experimentation (approval document No. 2017-1). Briefly, healthy adult male SD rats were acclimated in laboratory conditions, fed standard maintenance feed, and were provided sterilized water.

After adaptation, the rats weighed 150 ± 20 g and were randomly separated into six groups (*n* = 10 in each group), each group was fed a unique diet for 6 weeks and weighed every week. The normal control group was fed a normal diet and the other group was fed a high-fat diet (normal diet supplemented with 20% sucrose, 15% lard oil, 1.2% cholesterol, and 0.2% sodium cholate) to create models of rat obesity. After 1 week, different group rats were gavage-fed with different treat products. Each group was as follows: (1) normal control group (ND), normal diet + 0.5% CMC-Na; (2) model control group (HFD), high-fat diet + 0.5% CMC-Na; (3) positive control group (Orlistat), high-fat diet + 20 mg/kg orlistat; (4) high dose group (PFK-H), high-fat diet + 200 mg/kg phycobiliprotein, 40 mg/kg fucoxanthin, 400 mg/kg krill oil; (5) medium dose group (PFK-M), high-fat diet + 100 mg/kg phycobiliprotein, 20 mg/kg fucoxanthin, 200 mg/kg krill oil; (6) low dose group (PFK-L), high-fat diet + 50 mg/kg phycobiliprotein, 10 mg/kg fucoxanthin, 100 mg/kg krill oil. The rats’ weight and food intake were measured once per week during the 6 weeks treatment period [24].

The animals did not fast prior to blood collection; blood was collected and then sacrificed at the end of the experimental period.

### 2.4. Toxicity Analysis

The rats in three doses of lipid-lowering products fed 200 mg/kg phycobiliprotein, 40 mg/kg fucoxanthin, 400 mg/kg krill oil; 100 mg/kg phycobiliprotein, 20 mg/kg fucoxanthin, 200 mg/kg krill oil; 50 mg/kg phycobiliprotein, 10 mg/kg fucoxanthin, 100 mg/kg krill oil. Moreover, all the rats could access water at all times and were continuously monitored for mortality during the period of feeding.

### 2.5. Serum Biochemical Analysis

Blood was collected from the medial canthus of the eye or tail of the rats and the serum was isolated by centrifugation at 1200 rpm for 20 min. The serum was used for biochemical analysis. The collected serum samples were analyzed for activities of total cholesterol (TC), triglyceride (TG), high-density lipoprotein (HDL) cholesterol, and low-density lipoprotein (LDL) cholesterol levels with diagnostic kits. The liver tissue was homogenized in an ice bath to obtain suspension and estimated superoxide dismutase (SOD), malondialdehyde (MDA), acetyl-CoA carboxylase (ACC), fatty acid synthase (FAS), and glutathione (GSH) with diagnostic kits (Nanjing Jiancheng Corp., Nanjing, China). Then the Plasma atherogenic index and Castelli’s risk index were calculated by formula [25].
Plasma atherogenic index = lg (TG/HDL)
Castelli’s risk index (CRI-I) = TC/HDL
Castelli’s risk index (CRI-II) = LDL/HDL

### 2.6. Parameters to Analyze the Degree of Obesity

The food intake and weight changes of the rats were recorded every week. After blood collection, the organs including the liver, kidneys, and spleen were quickly removed and weighed. Besides, the perirenal and gonadal fat pads were harvested and weighed. Then, Lee’s index, adiposity index, liver index, kidney index, and spleen index were calculated by formula [26].
$$\mathrm{Lee}’s\;\mathrm{index}=\sqrt[{3}]{\mathrm{body}\;\mathrm{weight}}\;\times \;10^{3}/\mathrm{body}\;\mathrm{length}$$
Adiposity index = wet weight of body fat/body weight × 100
Liver index = wet weight of the liver/body weight × 100
Kidney index = wet weight of the kidney/body weight × 100
Spleen index = wet weight of the spleen/body weight × 100

### 2.7. Histopathological Analysis

Liver, kidney, and spleen tissues were fixed using 10% neutral formalin, embedded in paraffin. Paraffin-embedded tissues were sectioned and stained with hematoxylin and eosin and then were cut into slices of 5 μm. Examination with a microscope and observation of the organ sections of rats was conducted to analyze the tissue morphology and whether there are toxic side effects.

### 2.8. Statistic Statement

All data are expressed as means ± standard deviation (SD). Data were statistically analyzed using Microsoft Excel and SPSS 13.0. The images were drawn using Origin 2018. The data were analyzed by one-way ANOVA with LSD and Dunnett’s test and unpaired *t*-test. *p* < 0.05 (*, ^#^) indicated a statistically significant difference, and *p* < 0.01 (**) indicated an extremely statistically significant difference.

## 3. Results

### 3.1. Antioxidant Activities of PFK Complex In Vitro

The experiment compared the difference in antioxidant activities in single PBPs, FX, KO, and PFK complexes (Figure 1). The total antioxidant activity of the PFK complex had a statistically significant increase compared to the single PBPs or KO. The total antioxidant activity was almost a four-fold significant increase in comparison to FX. In addition, the hydroxyl-free radical scavenging activity showed similar results. The PFK complex increased significantly in hydroxyl-free radical scavenging activity as compared to the single products. The results indicated that the PFK complex showed more potent antioxidant activities compared to the single PBPs, FX, and KO. Therefore, the PFK complex was used for the subsequent in vivo experiments.

### 3.2. Effects of PFK Complex on Body Weight

Before the start of the experiment, every group of rats had similar body weights. The weight of high-fat diet rats was higher than normal diet rats after modeling. No significant differences were presented in terms of body weight among the groups except the normal control group, confirming that the high-fat model was successful. The body weights of all rats increased during the 6-week feed (Table 1). The PFK-L group’s average body weight was 370.1 ± 34.9 g and gained 73.0 ± 23.0 g after 6 weeks, which was lower than the HFD group (381.4 ± 28.9 g). The low dose of the PFK complex was able to significantly decrease body weight gains. Moreover, the Lee index and organ index were calculated (Figure 2A–E). As well, the adiposity index of the PFK-L group was 2.62 ± 0.41, which was significantly lower (*p* < 0.05) than the HFD group (3.64 ± 0.78). The effect of a low dose of PFK complex was not significantly different from orlistat. Furthermore, the other index has little difference in the remaining groups compared with the HFD group. Six weeks of feeding yielded no differences in average weekly consumption, which showed that the PFK complex may not affect appetite to weight loss (Figure 2F).

### 3.3. Effects of PFK Complex on Serum Lipids

Prior to the fed lipid-lowering products, high-fat diet groups (*p* < 0.05) had significantly higher serum TC levels, TG levels, and LDL-C levels than the ND group, which showed that obesity may lead to dyslipidemia. The LDL-C levels of the high-fat diet groups were slightly higher than the ND group. With lipid-lowering products, the serum TC levels, TG levels, and LDL-C levels were significantly lower in the PFK-L groups (*p* < 0.05) compared to the HFD group from the second week (Figure 3). At the end of the experiment, the serum TG levels of the PFK-L group and PFK-M group had significant differences with the HFD group, which were 22.90% and 32.11% lower, respectively. The serum TC level of the PFK-L group was 1.78 ± 0.22 mmol/L, which was lower than the Orlistat group (2.03 ± 0.39 mmol/L) and much lower than the HFD group (2.17 ± 0.33 mmol/L). The LDL-C has the same trend that the rats feeding on a low dose of PFK complex had significantly decreased LDL-C levels in the HFD group. The other group’s HDL-C levels had a small increase compared with the ND group with no significant differences. The plasma atherogenic index, Castelli’s risk index (CRI-I), and Castelli’s risk index (CRI-II) were associated with a high-fat diet [27]; the plasma atherogenic index, Castelli’s risk index (CRI-I), and Castelli’s risk index (CRI-II) were higher for rats in the HFD group compared with the ND group (Table 2). The plasma atherogenic index and Castelli’s risk index (CRI-II) in the PFK-L group had decreased to similar levels compared to the ND group indicating that they may reduce the risk of developing arterial stiffness. The PFK-H group may have more krill oil, with an increased liquid intake affecting the efficacy.

### 3.4. Effects of PFK Complex on Lipid Metabolism

The level of acetyl-CoA carboxylase (ACC) and fatty acid synthase (FAS) in the HFD group were significantly increased compared with the ND group (*p* < 0.05). There was a reduction with the treatment of orlistat and PFK complex (Figure 4A,B). The ACC and FAS were 0.18 ± 0.05 ng/mL and 8.67 ± 0.41 ng/mL with the PFK-L group producing an effect comparable to that of the PFK-H group and PFK-M group (*p* < 0.05), which was slightly lower than the Orlistat group. It demonstrated that the PFK complex had a good effect on the decrease of blood lipids, according to the guidance method for the evaluation of weight loss function by the China Food and Drug Administration.

### 3.5. Effects of PFK Complex on Antioxidant Status

The antioxidant status was evaluated by superoxide dismutase (SOD), malondialdehyde (MDA), and glutathione (GSH) (Figure 4C–E) [28]. It was evident that the content of MDA in HFD group rats was 0.35 μmol/g, which was significantly higher than the ND group rats (0.18 μmol/g) (*p* < 0.05) and the content of SOD and the enzyme activity of SOD in the HFD group rats were lower than ND group rats indicating that severe oxidative stress and lipid peroxidation occurred from a high-fat diet. The MDA and GSH decreased to 0.28 μmol/g and 4.26 g/L compared with HFD group rats after administration of a low dose of PFK complex over six weeks. In comparison with the HFD group, the other groups with different doses and orlistat also reduced the MDA and grew the SOD, manifesting that the antioxidant activity of obese rats could reach normal levels or even higher through a supplement of PFK complex.

### 3.6. Histopathological Observations

Histopathological observations of the liver, kidney, and spleen were carried out and the results are shown in Figure 5. The cells swelled and the cell boundary was blurry and had lipid droplets, which indicated the HFD rat’s liver had severe hepatic damage, compared with the ND group cell with no steatosis (Figure 5A). The damages to livers were recovered after treatment with a low dose of PFK complex. They decreased the gaps of cells and the number of lipid droplets, demonstrating that the PFK complex was able to promote anti-obesity. Similarly, the high-fat diet caused damage to the kidney and spleen. In HFD group rats, the expansion of the glomerular interstitial, the tubular epithelial vacuoles in the kidney, and the lipid vacuoles in the spleen were formed (Figure 5B,C). With the treatment low dose of PFK complex, these pathological changes were resolved, showing that the low dose of PFK complex had the potential to protect the organs in high-fat diets.

### 3.7. Toxicity Analysis

It was observed that the rats treated with different doses of PFK complex did not show any general appearance or behavioral changes or toxic symptoms. In addition, PFK complexes were practically non-toxic products due to no deaths occurring until the experiment was complete.

## 4. Discussion

Marine activity products from marine organisms such as algae, fish, and crustaceans, are attracting a lot of attention as new medicines and functional supplements are decreasing the risk of disease. Many active marine products have been confirmed to be safe and effective in the treatment of obesity [17,29]. In this study, we researched how the PFK complex could reduce body weight, lipid disorder, and risk of arteriosclerosis in 6 weeks of treatment. The results proved that the PFK complex appeared to be highly effective in weight loss and lipid-lowering, especially in decreasing the TG and LDL-C, which were based on a guidance way for the evaluation of weight loss function by the China Food and Drug Administration. Furthermore, the PFK complex has an effect on improving the antioxidant activity of the body. These results suggest that the PFK complex may be a promising treatment agent for obesity and associated metabolic diseases.

Previously, KO has been reported to have effects on lipid-lowering in the literature and fucoxanthin has weight loss effects [30]. KO is rich in phospholipids and omega-3 long-chain polyunsaturated fatty acids proven to promote cholesterol efflux to improve blood lipids [31]. Moreover, fucoxanthin promotes lipolysis by activating the UCP1 protein to achieve lipid-lowering effects [32,33]. In this study, we found that the low dose of the PFK complex exerted a hypolipidemic effect and a reduction of the damage to the liver, kidney, and spleen due to obesity in the model rats. The Lee index and adiposity index were 305.61 and 2.62 in low doses of PFK complex treatment, which demonstrated that the low dose of PFK complex not only had an effect on weight loss but also had a hypolipidemic effect. High and middle doses of the PFK complex have more krill oil, which is a liquid that may cause fat accumulation and restrict the effects of weight loss. Low doses of the PFK complex simultaneously realized weight loss and lipid-lowering goals. The TG, TC, and LDL-C tended to drop and the HDL-C tended to dose in PFK complex treatment, compared with HFD group rats. The HDL-C accelerates cholesterol catabolism and effluxes protected against atherosclerosis [34]. The high levels of TC and LDL-C easily result in cholesterol deposits on blood vessel walls and a high risk of atherosclerosis [35]. The plasma atherogenic index, Castelli’s risk index (CRI-I), and Castelli’s risk index in the experiment confirmed this. The unfavorable outcome of the high dose of PFK complex, which may relate to the KO, excessive intake may disturb the balance of lipid metabolism influencing the effects of obesity treatment.

Obesity is essentially caused by the imbalance between intake and consumption [36]. Anti-obesity can be achieved not only through increased lipid consumption but also by restricting adipogenesis. ACC and FAS are important rate-limiting enzymes in catalyzing the synthesis of fatty acids [37]. Yan reported that the inhibitors of FAS reduced lipid accumulation and increased Mal-CoA, Ac-CoA, and NADPH release, which reduced fat accumulation, inhibited adipocyte differentiation, and expedited energy consumption to treat obesity [38]. The de novo synthesis of lipids was slowed down with the suppression of ACC, but some inhibitors of ACC lead to the elevation in serum TG levels. In the present work, the PFK complex resulted in a two-third reduction in the ACC and one-third of the FAS when compared with the HFD group and did not elevate the TG, which showed that the PFK complex may be an inhibitor of ACC and FAS to fulfill weight loss, lipid-lowering, and protection of multiple organs, including liver, kidney, and spleen without side effects in contrast to chemical drugs [39].

There is growing literature suggesting that adiposity is reduced through the induction of multiple antioxidant enzymes and the activation of antioxidant enzymes [40]. The PBPs, KO, and FX have antioxidant activity [41,42]. The PFK complex has a superior antioxidant activity to single PBPs, KO, and FX in vitro experiments. MDA is an indicator of oxidative damage in vivo and can show the health status of people. High-fat diet caused high levels of MDA and hepatic and renal impairment. With lipid peroxidation, the MDA is generated, which limits the LDL-C transport and has cytotoxicity, increasing the risk of dyslipidemia and cardiovascular issues [43]. In the present work, the PFK complex enhanced the antioxidant enzyme activities and GSH to alleviate the MDA levels and lipid peroxidation and promote the tricarboxylic acid cycle. This appeared to play a synergistic role in anti-obesity.

The histopathological observations of the liver, kidney, and spleen confirmed that the analysis of biochemical parameters and the PFK complex has no side effects. This evidence has shown that the low dose of PFK complex had more biological efficacies and good abilities to reduce organ damage and weight loss and lipid-lowering in a high-fat diet compared with other doses in experiments. These provide a basis for anti-obesity with natural products other than therapeutic agents and support more deep study on this study.

## 5. Conclusions

In this study, we evaluated the anti-obesity, lipid-lowering, and antioxidant activities of the different doses of the PFK complex. The synergistic complex of PFK exhibited various efficacies including controlling body weight, reducing the levels of serum lipid markers, and reducing damage to organs from a high-fat diet. Too high a dose of PFK complex increased the intake of lipids, influencing the effects. One of the mechanisms for this effect was to regulate the adipogenesis of lipid and lipid peroxidation by limiting lipid synthesis enzyme activities and increasing antioxidant activity. These results suggested that the low dose of PFK complex could be a natural and healthy food to relieve obesity and hyperlipidemia, valuable for developing medicines and clinical trials.

## Figures and Tables

**Figure 1 nutrients-14-04815-f001:**
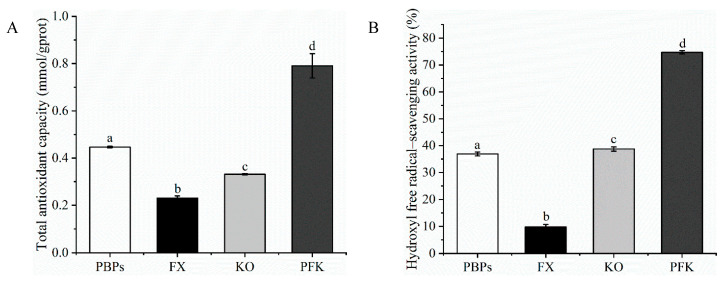
Antioxidant activities in vitro. (**A**) The total antioxidant activity. (**B**) Hydroxyl-free radical scavenging. Different superscript letters (a, b, c, d) within the same biochemical parameter show significant difference (*p* < 0.05).

**Figure 2 nutrients-14-04815-f002:**
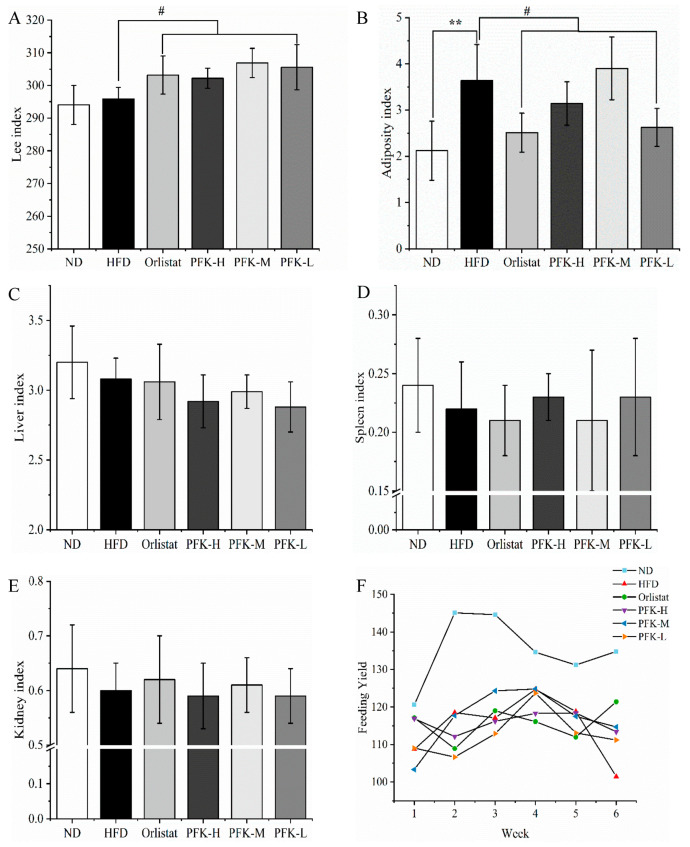
Effects of PFK on body index. (**A**) Lee’s index, (**B**) adiposity index, (**C**) liver index, (**D**) spleen index, (**E**) kidney index, and (**F**) feeding yield. ND group: normal diet + 0.5% CMC-Na; HFD group: high-fat diet + 0.5% CMC-Na; Orlistat group: high-fat diet + 20 mg/kg orlistat; PFK-H group: high-fat diet + 400 mg/kg krill oil, 40 mg/kg fucoxanthin, 200 mg/kg phycobiliprotein; PFK-M group: high-fat diet + 200 mg/kg krill oil, 20 mg/kg fucoxanthin, 100 mg/kg phycobiliprotein; PFK-L group: high-fat diet + 100 mg/kg krill oil, 10 mg/kg fucoxanthin, 50 mg/kg phycobiliprotein. The values were reported as the mean ± SD of rats per group. ** *p* < 0.01, ^#^ *p* < 0.05.

**Figure 3 nutrients-14-04815-f003:**
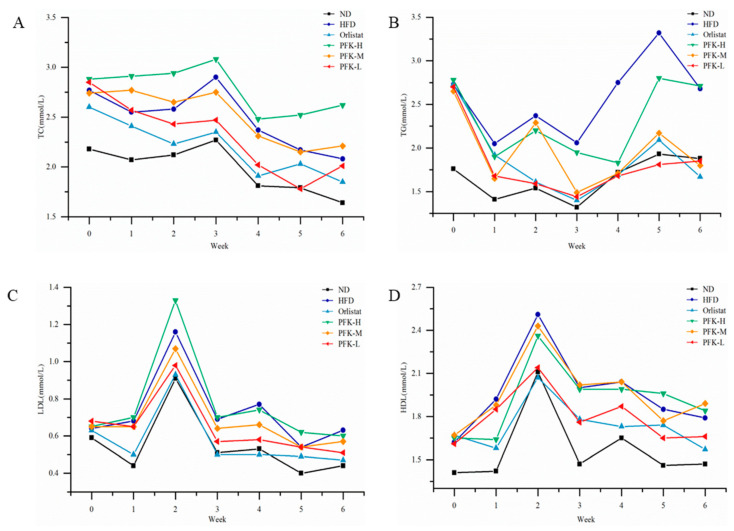
Effects of PFK on serum lipids. (**A**) TC, (**B**) TG, (**C**) LDL-C, (**D**) HDL-C. ND group: normal diet + 0.5% CMC-Na; HFD group: high-fat diet + 0.5% CMC-Na; Orlistat group: high-fat diet + 20 mg/kg orlistat; PFK-H group: high-fat diet + 400 mg/kg krill oil, 40 mg/kg fucoxanthin, 200 mg/kg phycobiliprotein; PFK-M group: high-fat diet + 200 mg/kg krill oil, 20 mg/kg fucoxanthin, 100 mg/kg phycobiliprotein; PFK-L group: high-fat diet + 100 mg/kg krill oil, 10 mg/kg fucoxanthin, 50 mg/kg phycobiliprotein.

**Figure 4 nutrients-14-04815-f004:**
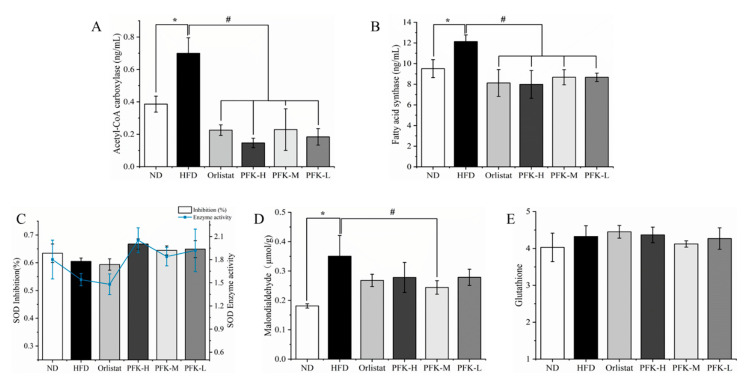
Effects of PFK on lipid metabolism and antioxidant activity. (**A**) ACC, (**B**) FAS, (**C**) SOD, (**D**) MDA, (**E**) GSH. ND group: normal diet + 0.5% CMC-Na; HFD group: high-fat diet + 0.5% CMC-Na; Orlistat group: high-fat diet + 20 mg/kg orlistat; PFK-H group: high-fat diet + 400 mg/kg krill oil, 40 mg/kg fucoxanthin, 200 mg/kg phycobiliprotein; PFK-M group: high-fat diet + 200 mg/kg krill oil, 20 mg/kg fucoxanthin, 100 mg/kg phycobiliprotein; PFK-L group: high-fat diet + 100 mg/kg krill oil, 10 mg/kg fucoxanthin, 50 mg/kg phycobiliprotein. The values were reported as the mean ± SD of rats per group. * *p* < 0.05, ^#^ *p* < 0.05.

**Figure 5 nutrients-14-04815-f005:**
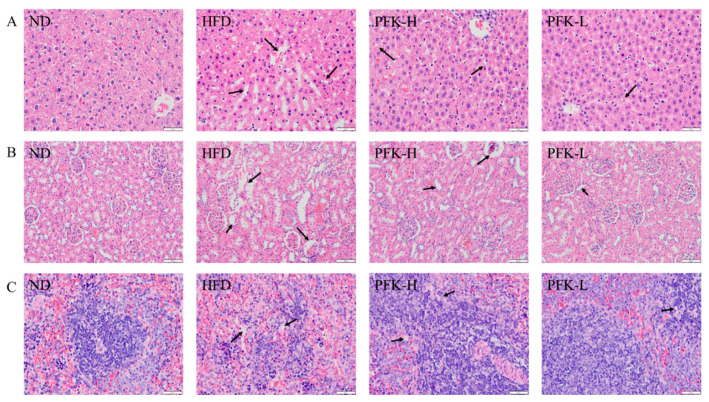
H and E staining of tissue. (**A**) Liver, (**B**) kidney, (**C**) spleen. ND group: normal diet + 0.5% CMC-Na; HFD group: high-fat diet + 0.5% CMC-Na; PFK-H group: high-fat diet + 400 mg/kg krill oil, 40 mg/kg fucoxanthin, 200 mg/kg phycobiliprotein; PFK-L group: high-fat diet + 100 mg/kg krill oil, 10 mg/kg fucoxanthin, 50 mg/kg phycobiliprotein.

**Table 1 nutrients-14-04815-t001:** Effects of PFK complex on body weight.

	Weight before Modeling/g	Weight before Administration/g	Weight after 6 Weeks/g	Weight Change/g
ND	169.4 ± 13.2	282.9 ± 22.8	370.1 ± 37.6	87.2 ± 29.6 ^a^
HFD	171.1 ± 13.2	296.1 ± 20.2	381.4 ± 28.9	85.3 ± 15.2 ^a^
Orlistat	170.0 ± 9.8	291.1 ± 27.7	360.6 ± 41.8	69.5 ± 24.8 ^b^
PFK-H	168.6 ± 5.9	298.8 ± 28.9	382.3 ± 29.0	89.8 ± 17.5 ^a^
PFK-M	167.8 ± 10.8	298.0 ± 32.2	379.1 ± 35.8	81.1 ± 28.9 ^a^
PFK-L	169.3 ± 10.6	297.1 ± 23.3	370.1 ± 34.9	73.0 ± 23.0 ^c^

^a^ *p* < 0.05, ^b^ *p* < 0.05, ^c^ *p* < 0.05.

**Table 2 nutrients-14-04815-t002:** Effects of PFK complex on risk index.

	Plasma Atherogenic Index	Castelli’s Risk Index (CRI-I)	Castelli’s Risk Index (CRI-II)
ND	0.10684	1.11565	0.29932
HFD	0.17528	1.16201	0.35196
Orlistat	0.02682	1.17834	0.29936
PFK-H	0.16815	1.42391	0.32609
PFK-M	−0.02119	1.16931	0.30159
PFK-L	0.04706	1.21084	0.30723

## Data Availability

Not applicable.

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
