# Peer review of "The Complex of Phycobiliproteins, Fucoxanthin, and Krill Oil Ameliorates Obesity through Modulation of Lipid Metabolism and Antioxidants in Obese Rats"

_nutrients, 2022, doi:10.3390/nu14224815_

Round 1

Reviewer 1 Report

The authors study the impact of supplementation of phycobiliproteins, fucoxanthin and krill oil (PFK) on the adverse effects of a high fat diet. They find PFK can correct adverse affects on body composition, lipid profiles, oxidative damage, and organ histology.

Major comments:

Figure 2B: Why is p reported to be <0.05 between HFD and PFK-M groups when they are nearly identical in adiposity index? Also, why would there be no effect in the PFK-M group when PFK-H and PFK-L are both impacted?

Table1: What is "modeling"?

Line 217: "Furthermore, the other index were decreased in the remaining 217 groups compared with HFD group, the decrease was not significant."
The statement is misleading. One cannot say two measurements are different if that difference is not found to be significant by a statistical test. We use statistical tests to determine if measurements are different.

Tables 2-5 would be better visualized as time-course plots.

Line 247: "The Plasma atherogenic index, 247 Castelli’s risk index (CRI-I) and Castelli’s risk index (CRI-II) were decreased almost simi- 248 lar with ND group by feeding PFK complex, indicating that they may reduce the risk of 249 developing arterial stiffness."
This statement is not true. There are instances where the indices are elevated by PFK feeding. Also, some of the indices where there is a reduction due to PFK feeding, the levels are closer to those of HFD than ND.

Section 3.4: In the discussion of dietary supplements impacting protein expression, the authors should site Jeong and Vacanti, Nutrition and Metabolism 2020 "Systemic vitamin intake impacting tissue proteomes".

The authors should address why dose-dependent impacts of PFK were observed in the Conclusions section.

Reviewer 2 Report

This paper is correct and suitable for the publication
